# Improving drug–target affinity prediction by adaptive self-supervised learning

Qing Ye[1] and Yaxin Sun[2,3]

[1] School of Data Science and Artificial Intelligence, Wenzhou University of Technology, Wenzhou, China
[2] School of Computer Science and Technology (School of Artificial Intelligence), Zhejiang Normal University, Jinhua, China
[3] Department of Algorithm, Zhejiang Aerospace Hengjia Data Technology Co. Ltd., Jiaxing, China



## ABSTRACT

Computational drug-target affinity prediction is important for drug screening and discovery. Currently, self-supervised learning methods face two major challenges in drug-target affinity prediction. The first difficulty lies in the phenomenon of sample mismatch: self-supervised learning processes drug and target samples independently, while actual prediction requires the integration of drug-target pairs. Another challenge is the mismatch between the broadness of self-supervised learning objectives and the precision of biological mechanisms of drug-target affinity (*i.e.*, the induced-fit principle). The former focuses on global feature extraction, while the latter emphasizes the importance of local precise matching. To address these issues, an adaptive self-supervised learning-based drug-target affinity prediction (ASSLDTA) was designed. ASSLDTA integrates a novel adaptive self-supervised learning (ASSL) module with a high-level feature learning network to extract the feature. The ASSL leverages a large amount of unlabeled training data to effectively capture low-level features of drugs and targets. Its goal is to maximize the retention of original feature information, thereby bridging the objective gap between self-supervised learning and drug-target affinity prediction and alleviating the sample mismatch problem. The high-level feature learning network, on the other hand, focuses on extracting effective high-level features for affinity prediction through a small amount of labeled data. Through this two-stage feature extraction design, each stage undertakes specific tasks, fully leveraging the advantages of each model while efficiently integrating information from different data sources, providing a more accurate and comprehensive solution for drug-target affinity prediction. In our experiments, ASSLDTA is much better than other deep methods, and the result of ASSLDTA is significantly increased by learning adaptive self-supervised learning-based features, which validates the effectiveness of our ASSLDTA.

## INTRODUCTION

The primary goal of drug-target affinity prediction (DTAP) is to predict the binding affinity between a drug and its target protein, which determines the therapeutic effect and safety of the drug. However, DTAP through experimental methods is often challenging

Corresponding author
Yaxin Sun,
sunyaxin2005@foxmail.com

and time-consuming. With the advancement of deep learning, introducing deep learning into DTAP has become a hot research topic (*Wang et al., 2024*), which can significantly accelerate drug development and reduce resource consumption.

Convolutional neural networks (CNN) can extract features from local windows. Deep drug-target affinity (DTA) is a well-known CNN-based method, whose drugs and targets are represented as matrixes by label encoding and embedding (*Öztürk, Özgür & Ozkirimli, 2018*). Based on the deep DTA, many improved methods have been proposed. Firstly, the 1D CNN in deep DTA has been replaced by other CNN structures, such as Dilated Gated CNN (*Zhu et al., 2023b*), and ResNet (*Zhang et al., 2024*), residual gated CNN (*Zhao et al., 2024*), and 2D CNN (*Han, Kang & Guo, 2024*). These CNN structures could extract more complex features. Secondly, more features have been inputted into CNN, such as pocket (*Yang et al., 2024; Jin et al., 2023*), atom coordinates and atom types (*Wang, Wu & Wang, 2024*), the output of correlation of different biological sequences (*Hua et al., 2023*), pre-training features (*Kalemati, Zamani Emani & Koohi, 2024*), and transform modules (*Wang et al., 2022*), which can provide different information for the CNN. Thirdly, the network parameters are initialized by transfer learning (*Tanoori, Zolghadri Jahromi & Mansoori, 2021*) or the stacked autoencoders (*Bahi & Batouche, 2021*), which can utilize a large number of unlabeled training samples to initialize the CNN parameter. Fourthly, CNN is used together with other networks, such as Transformer (*Li et al., 2024*), multi-head attention layer (*Chen et al., 2024*), and graph convolutional networks (*Deng et al., 2024; Öztürk, Ozkirimli & Özgür, 2019*).

Graph convolutional networks (GCN) can extract effective features from graph-structured data. Graph DTA is a well-known GCN-based method, whose drugs are represented as graphs (*Nguyen, Le & Venkatesh, 2021*). Based on graph DTA, several improved methods have been proposed. Firstly, attention mechanisms have been added to the GCN framework to obtain effective representations of the drug from different levels (*Tian et al., 2024; Wu et al., 2024*). Secondly, GCN has been used with CNN (*Nguyen, Le & Venkatesh, 2021*) or recurrent neural network (RNN) (*Wang et al., 2023; Mukherjee, Ghosh & Basuchowdhuri, 2022; Liu et al., 2015*) to extract more features. Thirdly, a cross-scale graph contrastive learning has been designed to combines features learned from the molecular scale and the network scale to capture information from both local and global perspectives (*Wang et al., 2024*), and the atomic-level protein pockets are taken as the input (*Lu et al., 2023*). Fourthly, GCN has been used to extract features from the target (*Xia et al., 2023a; Ye, Zhang & Lin, 2023*), which can increase the diversity of target features.

There are two problems with the CNN and GCN methods mentioned above. One problem is that the methods do not notice that DTAP is mainly determined by the core fragments of the drug and the target. In CNN, the convolution kernel can extract features for a local window, which can be viewed as a fragment of the input. However, the size of the convolution kernel cannot be too large, as a larger convolutional kernel implies more parameters, a greater number of kernels and a higher computational complexity (*Springenberg et al., 2014*). The fragment divided by CNN is insufficient to describe the core fragment. In GCN methods, graphs of GCN methods are created by the raw input but not by the fragment, which makes the graph global. Another problem is that the high-level

feature is directly extracted from raw input, which could be hard in DTAP, as the labeled training samples are too limited.

Self-supervised learning (SSL) can be trained by a large number of unlabeled samples, which can be used to overcome the problem of an insufficient training sample problem. Most SSL models are based on A Robustly Optimized BERT Pretraining Approach (RoBERTa) (*Liu et al., 2019*). GCN-BERT used two RoBERTa to extract features for the drug and target (*Lennox, Robertson & Devereux, 2021*). However, the result is not very good, where the CIs of the GCN-BERT on Kiba and Davis are 0.888 and 0.896 (*Lennox, Robertson & Devereux, 2021*). PortBERT (*Elnaggar et al., 2020*) and ProtALBERT (*Elnaggar et al., 2020*) are also used to extract features for the target (*Liu et al., 2021*). However, this method was not run on protein sequences longer than 1,000 (*Liu et al., 2021*) because of the limited resources of the computer. Furthermore, this method was only evaluated on a subset of Kiba. The sequence representations of the drug and the target are extracted by using the pretrained protein language model (*Zhao et al., 2024*), 300-dimensional pre-training features of the drug are used together with other features (*Wang & Li, 2023*). However, they are not evaluated on Kiba and Davis.

There are many reasons for the above phenomenon. Firstly, the output of RoBERTa may not meet the needs of DTAP. The DTA may be mainly determined by the substructure of the drug and target. However, the output of the above RoBERTa models (*Lennox, Robertson & Devereux, 2021*; *Liu et al., 2021*) is the pooling of inputs, whose core fragments could be overwhelmed. Secondly, the training goal of the RoBERTa is very different from that of the DTAP. However, the RoBERTa is directly used to extract high-level features. Thirdly, the input for text processing is very different from the input for DTAP, where RoBERTa was first designed for text processing. For example, a sequence of a target is much longer than a text sentence. The repetition probability of words in drugs and targets is much higher than that in text processing.

To overcome the above problems, in this article, an adaptive self-supervised learning-based drug–target affinity prediction is proposed, which contains a newly designed adaptive self-supervised learning training, two 2D CNN, a GCN, and a fully connected neural network.

In adaptive self-supervised learning, two RoBERTa models are trained on a large number of unlabeled fragments but not the whole sequences, which can overcome the input problem. Then, adaptive self-supervised learning can extract features with enough information to reconstruct fragments and their neighbor relationships, which can overcome the output problem. In the 2D CNN, the useful fragments and their neighbor relationships can be highlighted, and the negative positional information among fragments can be reduced, which can overcome the training goal problem. In the GCN, the feature of the graph structure of the drug can be extracted, as drug molecules have a typical graph structure. In the fully connected neural network, features extracted by two 2D CNN and a GCN can be integrated to predict DTA.

The contribution of this article can be concluded as follows: We designed a two-stages feature extraction method, which consists of an ASSL and a 2D CNN. The ASSL can learn enough information to reconstruct fragments and to describe the relationship among their

neighbors by unlabeled samples. The 2D CNN can further highlight the useful fragments and their neighbor relationships by a small number of labeled training samples. This method can bridge the objective gap between self-supervised learning and DTA prediction and alleviate the sample mismatch problem, and each stage undertakes specific tasks, fully leveraging the advantages of each model while efficiently integrating information from different data sources, providing a more accurate and comprehensive solution for drug-target affinity prediction.

# PROPOSED METHOD

## Data

The proposed model was first evaluated on five benchmark datasets of DTAP, namely, the Kiba (*Tang et al., 2014*), Davis (*Davis et al., 2011*), DTC (*Tang et al., 2018*), Metz (*Metz et al., 2011*), and Tox-Cast (*US Environmental Protection Agency, 2015*). The simple statistics for the sample information of these datasets are given in Table 1. It can be seen from Table 1 that there are only 2,111, 68, 5,983, 1,471, and 7,657 drugs and only 229, 442, 118, 170, and 328 targets on the above datasets. As a result, the prediction model could be hardly well trained only by these samples.

The proposed model was also evaluated on three benchmark binary classification datasets of compound–protein interaction (CPI) prediction, namely, the BindingDB (*Gao et al., 2018*), C.elegans (*Liu et al., 2015*), and Human (*Liu et al., 2015*) datasets. The simple statistics for the sample information of these datasets are given in Table 2. It can be seen from Table 2 that there are only 852, 1,767, and 1,696 drugs, and only 1,052 and 1,876 targets on the Human and C.elegans datasets. Although there are 53,253 targets on BindingDB, the used positive and negative samples are only 39,747 and 31,218. As a result, the problem of sample shortage still exists in CPI.

## Problems of SSL usd in DTAP and our solution strategy

SSL can utilize a large number of unlabeled samples to train the model, which can be used to overcome the problem of sample shortage, where RoBERTa is a typical SSL framework (*Liu et al., 2019*). RoBERTa was first used for text processing and got good results, which is trained with dynamic masking and removed the next sentence predicts loss (*Liu et al., 2019*). Because the simplified molecular input line entry system (SMILES) of the drug and the sequence of the target are similar to the sentence of the text, RoBERTa is considered to extract features for DTAP. However, RoBERTa cannot be directly used in DTAP, for the following reasons:

Firstly, the input of self-supervised learning is very different with the input of DTA. Self-supervised learning processes drug and target samples independently, while DTA prediction requires the integration of drug-target pairs. Obviously, this gap makes that the self-supervised learning hardly learns useful features for DTA prediction.

Secondly, the input is very different between text processing and DTAP. One difference is that the lengths of the SMILES and sequence are much longer than that of the sentence of the text. Simple statistics of lengths of SMILES and sequences are presented in Figs. 1 and 2, where the rectangle represents the interquartile range, top, median, and bottom

**Table 1 Simple statistics for the sample information of five DTA datasets.**

| Data sets | Drugs | Targets | Used drug-targets pairs | Affinity mean | Affinity STD |
|---|---|---|---|---|---|
| Kiba | 2,111 | 229 | 118,254 | 11.83 | 0.81 |
| Davis | 68 | 442 | 30,056 | 5.45 | 0.89 |
| DTC | 5,983 | 118 | 67,894 | 5.89 | 1.02 |
| Metz | 1,471 | 170 | 35,307 | 6.22 | 0.96 |
| Tox-Cast | 7,657 | 328 | 342,869 | 1.38 | 0.87 |

**Table 2 Simple statistics for the sample information of three CPI datasets.**

| Data sets | Drugs | Targets | Used drug-targets pairs |
|---|---|---|---|
| Human | 852 | 1,052 | 3,369(+)/3,359(−) |
| C.elegans | 1,767 | 1,876 | 3,893(+)/3,893(−) |
| BindingDB | 1,696 | 53,253 | 39,747(+)/31,218(−) |

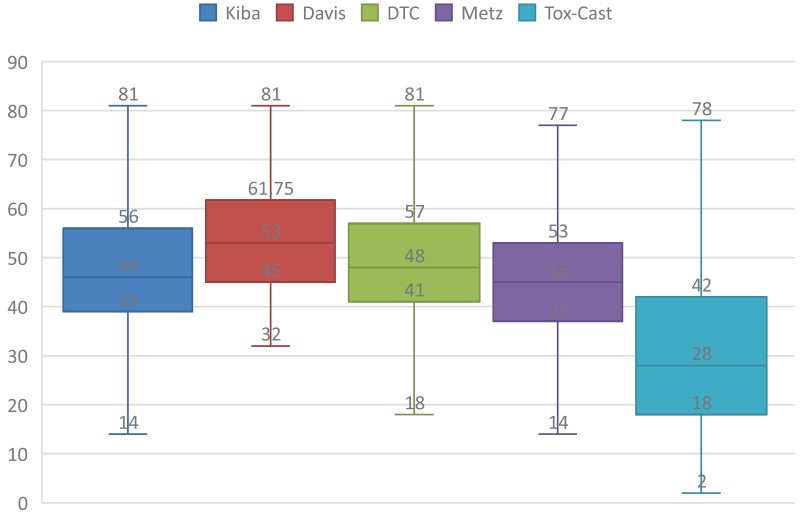

**Figure 1 Simple statistics for the length of the SMILES on five DTA datasets.**

lines represent the upper limit, median, and lower limit of the lengths. It can be seen from Fig. 1 that the median lengths of SMILESs are respectively about 46, 53, 48, 45, and 28. It can be seen from Fig. 2 that the median lengths of sequences are respectively about 620, 632, 673, 631, and 479. Each word could be represented by a long vector in RoBERTa, which makes that the dimension of the hidden state for a long sequence could be very high. Another problem is that the repetition probability of words in SMILES and sequence are much larger than that of the sentence, which makes the positional information very important in training the RoBERTa, and then outputs of similar sequences with different positions could be very different. As a result, the data distribution of DTAP is sparser.

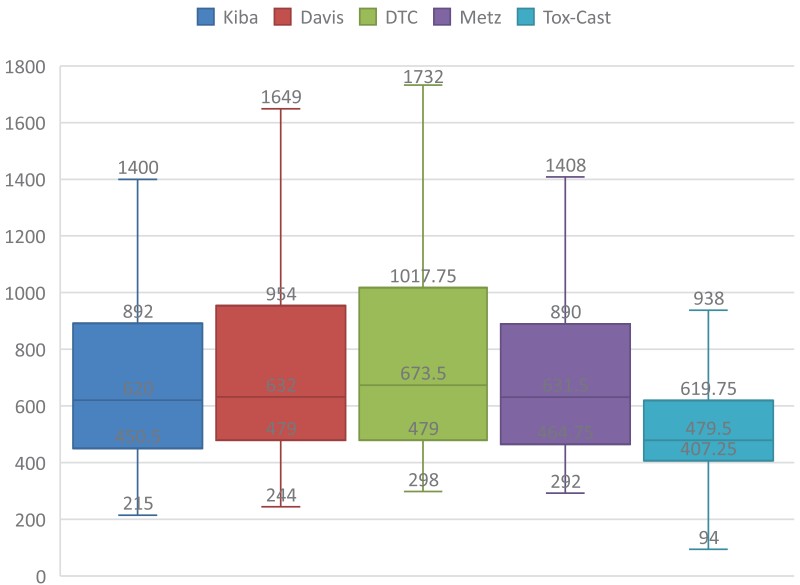

**Figure 2 Simple statistics for the length of the sequence on five DTA datasets.**

Thirdly, the output of the RoBERTa could be not meeting the needs of DTAP. The positional feature for a sentence could be important in text processing, but the positional feature for a SMILES or a sequence could be not very important in DTAP, as the DTAP is mainly determined by core fragments of the SMILES and sequence (*Lin, Zhang & Xu, 2020*; *Jin et al., 2023*), which is the principle of fragment-based drug design (*Shi et al., 2020*). An example can be shown by Fig. 3, where the core fragment of the drug is marked by a rectangle. It can be seen from Fig. 3 that the core fragment of the drug and the two core fragments of the target mainly determine the DTA. As a result, only the inner positional feature of the core fragment is important. Specifically, the positional feature between the core fragment and other fragments is bad for DTAP. Furthermore, if the pooling result of RoBERTa is used for DTAP, features of the core fragment could be overwhelmed. And if the whole result of RoBERTa is used, the dimension of features could be too high.

Fourthly, the training goal of the RoBERTa is very different from that of the DTAP. The training goal of the RoBERTa is to fill the dynamic masking. But the training goal of the DTAP is minimizing the mean square error between the prediction score and the actual score. Furthermore, because labeled training samples are too limited, the pre-trained RoBERTa is hardly fine-tuned by these labeled training samples. As a result, features extracted by RoBERTa should be further processed by a supervised method and cannot be directly used for DTAP. Specifically, features extracted by RoBERTa should own enough ability to describe the input and cannot lose much information.

To overcome the input problem, RoBERTa should be trained on fragments that are much shorter than the raw SMILES and sequence. To overcome the output problem, the output should be features of fragments and their relationship. To overcome the training

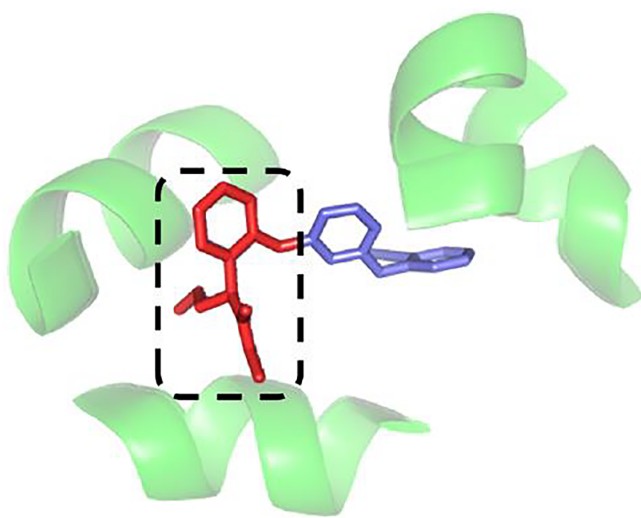

**Figure 3 An example used to show the core fragment (*Shi et al., 2020*).**

goal problem, RoBERTa should mainly extract low-level features, which should own enough ability to describe the input and cannot lose much information.

Based on the above analysis, an ASSL based on RoBERTa is designed to extract low-level features. In ASSL, RoBERTa is trained on a large number of unlabeled fragments, then the feature is extracted by RoBERTa on the adaptive sliding window. As a result, the extracted feature has enough ability to reconstruct the fragments and to describe the relationship among the fragments. Furthermore, the features extracted by ASSL are further processed by a 2D CNN to extract high-level features, as the ASSL is only used to extract low-level features, where the 2D CNN can highlight the useful fragments and their neighbor relationships, and reduce the negative positional features among fragments.

## ASSLDTA structure

The structure of ASSLDTA is shown in Fig. 4. It can be seen from Fig. 4 that ASSLDTA consists of five parts, such as ASSL, 2D CNN, GCN, and fully connected neural network (FCNN). In ASSL, low-level features of drugs and targets are extracted, which can be used to reconstruct fragments and their neighbor relationships. In the 2D CNN, the high-level features of drugs and targets are further extracted, which can highlight useful fragments and their neighbor relations. In GCN, graph-based high-level features of drugs are extracted since drug molecules have a typical graph structure. In FCNN, the high-level features extracted by two 2D CNNs and one GCN are integrated to predict the DTA. Here, it's worth noting that ASSL specifically focuses on extracting low-level features rather than directly tackling high-level ones, due to four identified issues with the current application of self-supervised learning (SSL) directly to DTA prediction.

## Features extracted by ASSL together with 2D CNN

ASSL consists of ASSL training and ASSL-based feature extraction. The ASSL training is first introduced. According to the motivation in "Problems of SSL usd in DTAP and Our

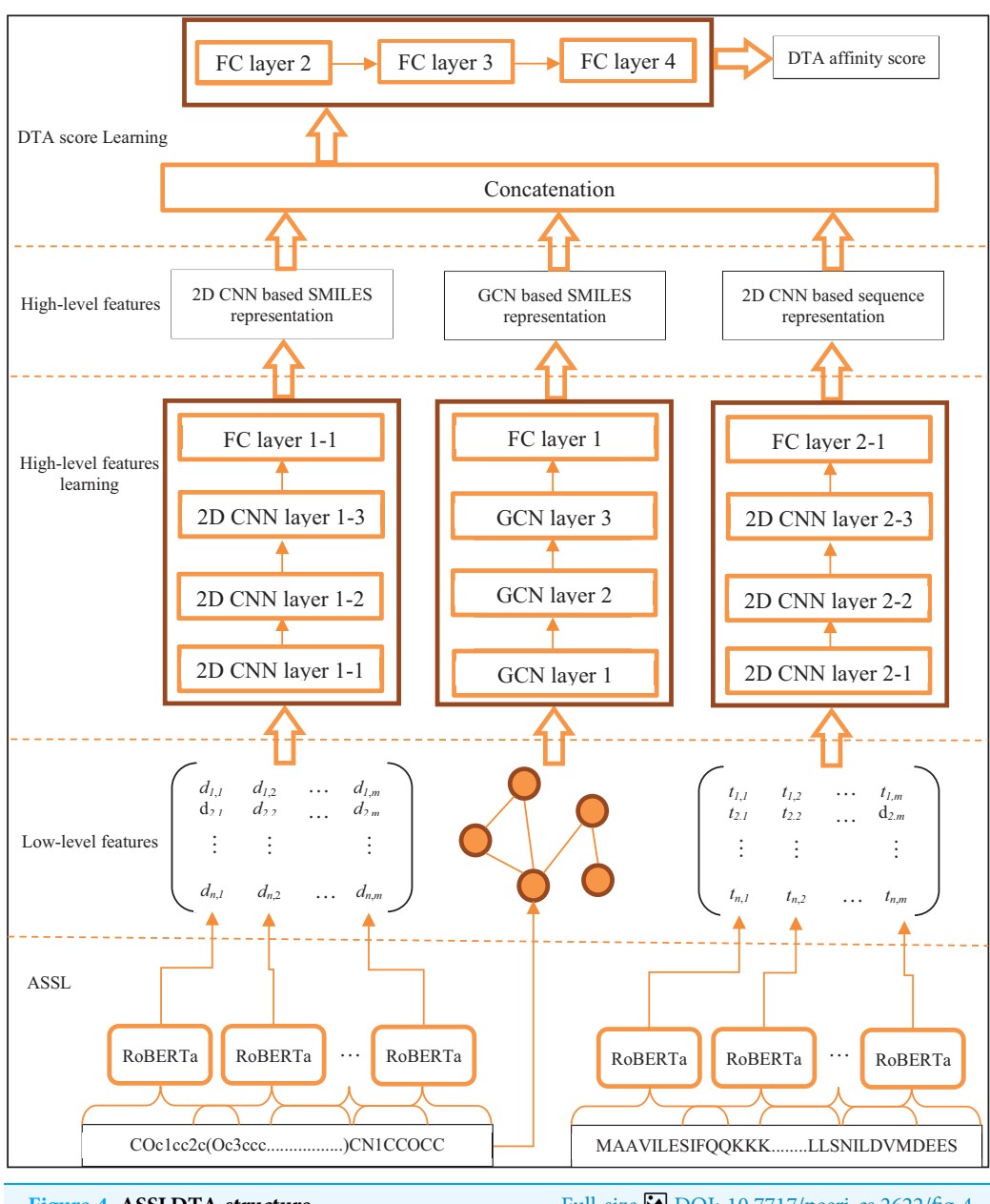

**Figure 4  ASSLDTA structure.**

Solution Strategy", the data prepared for RoBERTa should be the fragments of the drug or the target, and the lengths of fragments cannot be too large. However, it is still difficult to automatically divide appropriate fragments for the drug and target. Specifically, the lengths of automatically divided fragments may be very different, which also adversely affects the training of the model. As a result, fixed-length fragments of the SMILES and sequence are prepared for the RoBERTa. We opted for RoBERTa over BERT due to our implementation of a two-phase feature extraction approach. In the pre-training phase, our objective is to optimally preserve the intrinsic characteristics of individual fragment. Conversely, the interaction and relationship among these fragments, which constitute the inter-fragment

features, are intended to be developed and refined during the subsequent supervised training phase.

Given a training sequence $d_i = [d_{i,1}, d_{i,2}, \ldots, d_{i,pi}]$ whose length is $p_i$, the length of the fragment $l$, the $k$-th fragment of $d_i$ can be defined as:

$$s_i^k = [d_{i,k*l}, \ d_{i,k*l+1}, \ldots, d_{i,k*l+l-1}] \tag{1}$$

It can be seen from Eq. (1) that $d_i$ is divided into $p_i/l$ non-overlapping fragments. The adjacent fragments do not overlap, for the reason that the same word would be masked in one fragment but not masked in neighbor fragments when adjacent fragments overlap with each other.

After processing all training drugs by Eq. (1), the RoBERTa model $\widehat{f}_T$ of the drug can be trained by hugging face toolbox (Transformer, https://github.com/huggingface). The CHEMBL (*Gaulton et al., 2017*) dataset is used for training, which contains 2,105,464 drugs. Similarly, after processing all training targets by Eq. (1), the RoBERTa model $\widecheck{f}_T$ of the target also can be trained by the Hugging Face toolbox (Transformer, https://github.com/huggingface). The Swiss-Port (Swiss-Port dataset, http://www.gpmaw.com/html/swiss-prot.html) dataset is used for training, which contains 565,928 targets. Because $\widehat{f}_T$ and $\widecheck{f}_T$ are trained on the fragments, they have enough power to extract the features for the fragment.

After training the $\widehat{f}_T$ and $\widecheck{f}_T$, they can be used to extract the low-level features for the drug and target. The processing is named as ASSL based feature extraction. In the feature extraction, fixed-length fragments of the SMILES and sequence also should be prepared for the $\widehat{f}_T$ and $\widecheck{f}_T$.

Given a testing sequence $d_i = [d_{i,1}, d_{i,2}, \ldots, d_{i,pi}]$ whose length is $p_i$, the length of the fragment $l$, the sliding offset $\widehat{b}_i$, the $k$-th fragment of $d_i$ can be defined as:

$$s_i^k = [d_{i,k*bi}, \ d_{i,k*bi+1}, \ldots, d_{i,k*bi+l-1}] \tag{2}$$

It can be seen from Eq. (2) that $d_i$ is divided into $p_i/b_i$ overlap fragments by the adaptive sliding window. The reasons that the adjacent fragments overlap with each other are as follows. Firstly, it can increase the probability that the core fragment is only distributed in a fragment. Secondly, it can increase the variety of fragments, which can extract more features for the sequence. Thirdly, the fragment is not be dynamically masked in the feature extraction, so the overlap does not affect the model. Lastly, the number of fragments can be controlled by setting different $\widehat{b}_i$. This is important, as dimensions of inputs of many deep neural networks needs to be the same, such as the CNN and FCNN.

To make that numbers of fragments divided for different sequence are the same, $\widehat{b}_i$ is calculated by:

$$b_i = p_i/K \tag{3}$$

where $K$ is the number of fragments. It can be seen from Eqs. (2) and (3) that regardless of the length of a sequence, the number of fragments divided by Eq. (3) is $K$.

After processing a SMILES by Eq. (2), low-level features of this drug can be extracted by:

$$\widehat{O}_i = [\widehat{f}_T(\widehat{s}_i^1), \widehat{f}_T(\widehat{s}_i^2), \ldots, \widehat{f}_T(\widehat{s}_i^K)] \tag{4}$$

where $\widehat{s}_i^1, \widehat{s}_i^2, \ldots, \widehat{s}_i^K$ are fragments of $i$-th drug divided by Eq. (2), $\widehat{f}_T$ is the trained RoBERTa model of the drug, $\widehat{O}_i \in R^{u \times K}$ is the extracted low-level features of $i$-th drug, $\widehat{f}_T(\widehat{s}_i^k)$ is the output of $\widehat{f}_T$ whose dimension is $u \times 1$.

After processing a sequence of $j$-th target by Eq. (2), low-level features of this target can be extracted by:

$$\breve{O}_j = [\breve{f}_T(\breve{s}_j^1), \breve{f}_T(\breve{s}_j^2), \ldots, \breve{f}_T(\breve{s}_j^K)] \tag{5}$$

where $\breve{s}_j^1, \breve{s}_j^2, \ldots, \breve{s}_j^K$ are fragments of $j$-th target divided by Eq. (2), $\breve{f}_T$ is the trained RoBERTa model of the target, $\breve{O}_j \in R^{u \times K}$ is the extracted low-level features of $j$-th target.

Extracting features by Eqs. (4) and (5) has three advantages. Firstly, the extracted features own enough information to reconstruct the fragments. This is the property of the RoBERTa, as RoBERTa has a strong ability to fill the mask. Secondly, the extracted features own the relationships among fragments. As shown in Eqs. (4) and (5), $\widehat{O}_i$ and $\breve{O}_j$ contain features of all fragments and the positional information of these fragments. Thirdly, the extracted features are the low-level features. RoBERTa is only used to extract the features for fragments, and which fragments and what relationships among fragments benefit for DTAP are further processed in the next steps. Therefore, the problem that the training goal of RoBERTa is very different from that of the DTAP can be overcome.

$\widehat{O}_i$ and $\breve{O}_j$ are low-level features, so another deep neural network should be designed to further extract the high-level features. Because fragments are divided by an adaptive sliding window (see Eq. (2)), the core fragment could be distributed in several adjacent fragments. Therefore, the deep neural network should be able to extract relationships among adjacent fragments. Furthermore, features of a fragment extracted by RoBERTa may also need to be weighted. As a result, a 2D CNN is further used to extract the high-level features.

RoBERTa is only used to learn the low-level features. Firstly, RoBERTa is a pre-trained model that requires a large number of training samples, but DTAP has fewer training samples, making it difficult to fully train the model. Secondly, it is difficult to define suitable tokens for RoBERTa that are used to obtain high-level features because species of $\widehat{f}_T(\widehat{s}_i^2)$ and $\breve{f}_T(\breve{s}_j^K)$ may be infinite, which may be hardly used as the token.

Given the low-level features $\widehat{O}_i$ and $\breve{O}_j$ of the $i$-th drug and the $j$-th target, the high-level features can be extracted by:

$$\widehat{h}_i = \widehat{f}_C(\widehat{O}_i) \tag{6}$$

$$\breve{h}_j = \breve{f}_C(\breve{O}_j) \tag{7}$$

where $\widehat{f}_C$ and $\breve{f}_C$ are two 2D CNN, $\widehat{h}_i$ and $\breve{h}_j$ are 2D CNN-based high-level features of a drug and a target.

$\widehat{f}_C$ and $\breve{f}_C$ contain three 2D CNN layers. Each 2D CNN layer contains a 2D convolutional sublayer, a 2D batch normalization sublayer, and a rectified linear unit

activation function. The reasons that only three 2D CNN layers are used here are as follows: Firstly, 2D CNN is only used for learning high-level features from low-level features, so it does not need too many layers. Secondly, there are too few labeled training samples, and 2D CNN with too many layers is easy overfitting.

The features extracted by Eqs. (6) and (7) have three advantages. Firstly, core fragments can be highlighted by the convolution kernel, as the fragment benefits for DTAP can get a larger response value. Secondly, the relationship benefit for DTAP can be extracted by the convolution kernel, as the convolution kernel has a strong ability to extract local positional relationships. Thirdly, meaningless positional information among fragments can be reduced by 2D CNN, as the field of view is getting bigger and bigger from the lower level to the upper level.

## Features extracted by GCN and DTA score learned by FCNN

GCN is also used to extract high-level features for the drug. The reasons are as follows: Firstly, drug molecules have distinct graph structures, and the GCN can extract many useful features. Secondly, some special words are existed in SMILES, such as '.', '\', '/', '7', '8', '6', ')', '5' and '4', which may affect the effectiveness of RoBERTa.

Given a drug $d_i = [d_{i,1}, d_{i,2}, \ldots, d_{i,pi}]$ whose length is $p_i$, a global pooling GCN (*Zhang et al., 2021*) is used to learn GCN based high-level features for the drug, which can be calculated by:

$$\hat{h}_i = \hat{f}G(d_i) \tag{8}$$

where $\hat{f}_G$ is a GCN, which contains three GCN layers. For a GCN, the most crucial aspect is determining the graph vertices and edges. Similar to the approach in *Zhang et al. (2021)*, the vertices in the graph representing a drug molecule correspond to atoms, and the features of these vertices are represented using one-hot encoding. The edges of the graph for a drug molecule are formed based on its chemical bonds, which are obtained utilizing the RDKit tool (RDKit: open-source cheminformatics, https://www.rdkit.org). If a bond exists between two atoms, it signifies the presence of an edge; otherwise, no edge is formed. Once the graph is constructed, PyTorch Geometric (https://pytorch-geometric. readthedocs.io/en/latest/) can be employed to run the GCN.

After extracting high-level features for a drug and a target by Eqs. (6)–(8), features of this drug-target pair can be concatenated by $\widehat{h}_i$, $\breve{h}_i$ and $\hat{h}_i$ as follows:

$$h_i = [\widehat{h}_i, \breve{h}_i, \hat{h}_i]. \tag{9}$$

The compatibility of $\widehat{h}_i$, $\breve{h}_i$ and $\hat{h}_i$ in concatenating $h_i$ can be guaranteed for the following reasons. One is that the activation functions in 2D CNN and GCN can limit the output of neurons within a certain range. Another is that the last layer of our 2D CNN and GCN is a fully connected layer, which can adjust the output dimension.

The DTA score of the drug-target pair can be calculated as following:

$$a_i = f_D(h_i) \tag{10}$$

where $\hat{f}_D$ is a FCNN.

In the training, the DTA score of the drug-target pair is known, and $\hat{f}_C, \breve{f}_C, \hat{f}_G, f_D$ can be trained by the mean-square error between the prediction score and actual score.

## The proposed ASSLDTA model

ASSLDTA consists of the ASSL training, supervised DTAP training, and supervised DTAP testing. ASSL training consists of step 1 and step 2. Supervised DTAP training consists of steps 3 to 9, where steps 3 and 4 can be concluded as ASSL-based feature extraction, step 5 is the 2D CNN-based high-level feature extraction, step 6 is the GCN-based high-level feature extraction, steps 7 to 9 can be concluded as DTA score learned by FCNN. These have been introduced in detail in "Parametric Analysis of the ASSLDTA" and "Results of DTA on Kiba". Supervised DTAP testing consists of step 10 to step 15, which contain similar processes to supervised DTAP training. The code is available at https://github.com/yeqing0713/ASSLDTA.

In ASSLDTA, ASSL is used to learn the low-level features. Compared with other artificially defined low-level features, such as protein domains (*Wang, Wu & Wang, 2024*), motifs (*Wang, Wu & Wang, 2024*), recurrent neural network (RNN) (*Hua et al., 2023*), similarity matrixes (*Ye, Zhang & Lin, 2022*), 3D interaction information (*Han, Kang & Guo, 2024*), lots of traditional features (*Kalemati, Zamani Emani & Koohi, 2024*), and one-shot vector (*Zhang et al., 2021*), ASSL can utilize a large number of unlabeled training samples to learn more effective and comprehensive features. In ASSLDTA, two RoBERTa models are trained on fragments of drugs and targets and used to extract features for fragments. Compared with other pre-trained methods (*Liu et al., 2019*; *Lennox, Robertson & Devereux, 2021*; *Elnaggar et al., 2020*; *Liu et al., 2021*), ASSLDTA can overcome three problems of pre-trained methods mentioned in the "Problems of SSL usd in DTAP and Our Solution Strategy". In ASSLDTA, ASSL together with 2D CNN is utilized. Compared with supervision methods (*Öztürk, Özgür & Ozkirimli, 2018*; *Zeng et al., 2021*; *Wang, Wu & Wang, 2024*; *Aleb, 2021*), ASSLDTA can extract low-level features by a large number of unlabeled samples, and extract high-level features that highlight the core fragments and their relationships by limited labeled samples.

## Architectural parameter

ASSLDTA consists of two ASSL, two 2D CNN, a GCN, and an FCNN. Many architectural parameters exist in these neural networks, whose values are shown in Table 3.

ASSL primarily consists of RoBERTa. In ASSL, the number of attention heads and the number of hidden layers of RoBERTa are respectively set to 6 and 3, as the numbers of tokens are only 62 and 25 in the drug and target, which makes that a small RoBERTa model has enough power to extract features for these tokens. The hidden size of RoBERTa is set to 60, as this value must be a multiple of the number of attention heads. The pooler fully connected size is set to 60, which determines the dimension $u$ of $\hat{O}_i$ and $\breve{O}_j$. The $u$ cannot be too big, as setting a larger input size for 2D CNN usually requires deeper networks or larger convolution kernels, which could be overfitting when training samples are not enough. Batch size is set to 128, as our model is smaller than the default model and

Table 3 Architectural parameters settings of ASSLDTA.

| Model | Parameters | Values |
|---|---|---|
| RoBERTa | Max position embeddings | 512 |
| | Number of attention heads | 6 |
| | Number of hidden layers | 3 |
| | Pooler fully connected size | 60 |
| | Hidden size | 60 |
| | Mlm probability | 0.15 |
| | Batch size | 128 |
| 2D CNN | Number of filters | 64, 128, 256 |
| | Kernel size | 5, 3, 3 |
| | Kernel stride | 2 |
| | Max pool size | 3 |
| | FC layer 1-1 output | 1,024 |
| | FC layer 2-1 output | 2,048 |
| | Max pool stride | 2 |
| GCN | Hidden size | 78 |
| | Output dimension | 1,280 |
| | FC layer 1 output | 1,024 |
| | Dropout ratio | 0.1 |
| FCNN | Hidden size | 1,024, 512, 1 |
| | Dropout ratio | 0.1 |
| Global parameters | Batch size | 1,024 |
| | Learning rate (LR) | 0.001 |
| | Number of epoch | 2,000 |

can be trained by more samples in the same time. Other parameters are set to the default values in the Hugging Face toolbox (Transformer, https://github.com/huggingface). Since ASSL is a pre-trained model that requires extensive training time and is only used for extracting low-level features, the parameters have a limited impact on the experimental results within a wide range. Therefore, most of the ASSL parameters are set to fixed values based on the above analysis.

In the 2D CNN, the numbers of filters for three 2D CNN layers are respectively set to 64, 128, and 256. A total of 256 is set for the third 2D CNN layer, as the dimension of the output can be 2560, which is a bit larger than the dimension of the output of the FC layer 1-2. The kernel sizes are respectively set to 5, 3, and 3, as some useful features should be extracted in a larger field of view in the first layer. Dimensions of the output of FC layer 1-1 and FC layer 1-2 are set to 1,024 and 2,048, as the high-level features of the drug are extracted by two methods and that of the target is only extracted by a method.

The reasons of setting parameters for the GCN, FCNN, and global parameters are similar to that for the GCN and FCNN in *Zhang et al. (2021)*.

# EXPERIMENTS AND RESULTS

## Experimental setting

Five datasets listed in Table 1 are used to validate the method, such as Kiba, Davis, DTC, Metz, and Tox-Cast. The Human, C.elegans, and BindingDB datasets are used to validate the effectiveness of CPI. To verify generalization ability, our experiments split the dataset into training, validation, and test sets in a ratio of 8:1:1. Five-fold cross-validation was used, and the process was executed five times. The number of samples in the training, validation, and test sets for each dataset are presented in Table 4.

The concordance index (CI), mean squared error (MSE), $r_m^2$ and area under the precision-recall curve (AUPR) are used as the metrics for DTA, where CI measures the rank of the predicted binding affinity (*Mukherjee, Ghosh & Basuchowdhuri, 2022*), and MSE measures the difference between the vector of predicted values and the vector of the actual value (*Zhang et al., 2021*), $r_m^2$ calculated as the ratio of the sum of squared differences between the predicted values and the mean of the dependent variable to the total sum of squared differences between the actual values and the mean of the dependent variable (*Xia et al., 2023b*), AUPR measures the area under the curve that plots precision against recall (*Ye, Zhang & Lin, 2023*). The area under the ROC curve is used as the metric for CPI, where AUROC is the area under the receiver operating characteristic curve (*Ye, Zhang & Lin, 2022*). Precision is used as the metric for the fill masking of ASSL. Let $P$ and $N$ are the numbers of positive samples and negative samples of a dataset. $TP$ and $TN$ denote numbers of true positives and true negatives. $FP$ and $FN$ denote numbers of false positives and false negatives. Precision can be defined as:

$$\text{Precision} = \frac{TP}{TP + FP} \tag{11}$$

## Parametric analysis of the ASSLDTA

It can be seen from Algorithm 1 that three parameters should be set for ASSLDTA, such as $\hat{l}$, $\breve{l}$ and $K$. They are set according to the lengths of SMILESs and targets on five DTA datasets, which are presented in Figs. 1 and 2.

It can be seen from Fig. 1 that most lengths of SMILESs are distributed in 14 to 75, and lower quartile lengths of SMILESs are respectively 39, 45, 41, 37, and 18 on five DTA datasets. These lengths are not very larger than the lengths of sentences in the text. As a result, $\hat{l}$ should be set to the value that the fragment with this length contains enough information for DTA. In our experiment, $\hat{l}$ is set to 16, as 16 is slightly larger than 14, which makes that most core fragments can appear in one fragment and numbers of fragments divided for most SMILESs are not too big.

It can be seen from Fig. 2 that most lengths of sequences are distributed in 215 to 1,732, and lower quartile lengths of SMILESs are respectively 450, 479, 479, 464, and 407. These lengths are very larger than the lengths of sentences in the text. As a result, there are three factors to consider when setting $\breve{l}$, such as the lengths of the core fragments, the model processing power, and the number of fragments divided for most sequences. In our experiment, $\breve{l}$ is set to 128, which is slightly larger than the half of 215.

**Table 4 Simple statistics for the training, validation, and test sets for each dataset of eight datasets.**

| Data sets | Number of used drug-targets pairs | Number of training samples | Number of validation samples | Number of testing samples |
|---|---|---|---|---|
| Kiba | 118,254 | 94,604 | 11,825 | 11,825 |
| Davis | 30,056 | 24,044 | 3,006 | 3,006 |
| DTC | 67,894 | 54,316 | 6,789 | 6,789 |
| Metz | 35,307 | 28,243 | 3,532 | 3,532 |
| Tox-Cast | 342,869 | 274,296 | 34,287 | 34,286 |
| Human | 6,728 | 5,382 | 673 | 673 |
| C.elegans | 7,786 | 6,228 | 779 | 779 |
| BindingDB | 70,965 | 56,772 | 7,096 | 7,097 |

---

**Algorithm 1 Steps of ASSLDTA**

Steps of ASSLDTA

**Input:** Pre-training drugs $\widehat{D} = [\widehat{d}_1, \widehat{d}_2, \ldots, \widehat{d}_n]^T$, pre-training targets $\breve{T} = [\breve{t}_1, \breve{t}_2, \ldots, \breve{t}_m]^T$, training drugs $D = [d_1, d_2, \ldots, d_n]^T$, training targets
$T = [t_1, t_2, \ldots, t_m]^T$, the DTA score $Y$, the length of the fragment of the drug $\widehat{l}$, the length of the fragment of the target $\breve{l}$, the number of fragments $K$,
a test drug $d$ and a test target $t$.

**Output:** the predicted DTA score of $d$ and $t$.

**Steps:**

**Stage 1: ASSL training**

1: Divided fragments for $\widehat{D} = [\widehat{d}_1, \widehat{d}_2, \ldots, \widehat{d}_n]^T$ and $\breve{T} = [\breve{t}_1, \breve{t}_2, \ldots, \breve{t}_m]^T$ by $s_i^k = [d_{i,k*l}, \ d_{i,k*l+1}, \ldots, d_{i,k*l+l-1}]$.

2: Trained drug RoBERTa $\widehat{f}_T$ by drug fragments and trained target RoBERTa $\breve{f}_T$ by target fragments.

**Stage 2: supervised DTAP training**

3:    Divided drug fragments and target fragments for $D = [d_1, d_2, \ldots, d_n]^T$ and $T = [t_1, t_2, \ldots, t_m]^T$ by $s_i^k = [d_{i,k*bi}, \ d_{i,k*bi+1}, \ldots, d_{i,k*bi+l-1}]$.

4:    Extracted low-level features for drugs and targets by $\widehat{O}_i = [\widehat{f}_T(\widehat{s}_i^1), \widehat{f}_T(\widehat{s}_i^2), \ldots, \widehat{f}_T(\widehat{s}_i^K)]$ and $\breve{O}_j = [\breve{f}_T(\breve{s}_j^1), \breve{f}_T(\breve{s}_j^2), \ldots, \breve{f}_T(\breve{s}_j^K)]$.

5: Extracted 2D CNN-based high-level features for drugs and targets by $\widehat{h}_i = \widehat{f}_C(\widehat{O}_i)$ and $\breve{h}_j = \breve{f}_C(\breve{O}_j)$.

6: Extracted GCN based high-level features for drugs by $\hat{h}_i = \hat{f}_G(d_i)$.

7: Calculated the drug-target pair features by $h_i = [\widehat{h}_i, \breve{h}_i, \hat{h}_i]$.

8: Calculated the DTA scores $A$ by $a_i = f_D(h_i)$.

9: Training the $\widehat{f}_C, \breve{f}_C, \hat{f}_G, f_D$ by minimizing the mean-square error between $A$ and $Y$.

**Stage 3: supervised DTAP testing**

10: Divided drug fragments and target fragments for $d$ and $t$ by $s_i^k = [d_{i,k*bi}, \ d_{i,k*bi+1}, \ldots, d_{i,k*bi+l-1}]$.

11: Extracted low-level features for $d$ and $t$ by $\widehat{O}_i = [\widehat{f}_T(\widehat{s}_i^1), \widehat{f}_T(\widehat{s}_i^2), \ldots, \widehat{f}_T(\widehat{s}_i^K)]$ and $\breve{O}_j = [\breve{f}_T(\breve{s}_j^1), \breve{f}_T(\breve{s}_j^2), \ldots, \breve{f}_T(\breve{s}_j^K)]$.

12: Extracted 2D CNN-based high-level features for $d$ and $t$ by $\widehat{h}_i = \widehat{f}_C(\widehat{O}_i)$ and $\breve{h}_j = \breve{f}_C(\breve{O}_j)$.

13: Extracted GCN based high-level features for $d$ by $\hat{h}_i = \hat{f}_G(d_i)$.

14: Calculated the drug-target pair features by $h_i = [\widehat{h}_i, \breve{h}_i, \hat{h}_i]$.

15: Calculated the DTA scores $a$ by $a_i = f_D(h_i)$.

**Table 5  Results of ASSLDTA by setting 16, 24, and 32 for K on Kiba and Davis.** The bold indicates the best results of ASSLDTA when K takes different values.

| Metrics | K | Kiba | Davis | Mean |
|---|---|---|---|---|
| CI | 16 | 0.907 (0.003) | **0.910 (0.002)** | **0.9085** |
| | 24 | **0.908 (0.002)** | **0.910 (0.002)** | **0.9090** |
| | 32 | 0.906 (0.003) | 0.909 (0.003) | 0.9075 |
| MSE | 16 | 0.124 (0.002) | **0.196 (0.003)** | **0.1600** |
| | 24 | **0.123 (0.002)** | 0.197 (0.002) | 0.1600 |
| | 32 | 0.124 (0.002) | 0.199 (0.003) | 0.1615 |
| $r_m^2\uparrow$ | 16 | **0.786 (0.005)** | **0.758 (0.004)** | **0.7720** |
| | 24 | 0.785 (0.004) | **0.758 (0.004)** | 0.7715 |
| | 32 | 0.783 (0.005) | 0.755 (0.005) | 0.7690 |
| AUPR↑ | 16 | **0.846 (0.004)** | 0.802 (0.005) | 0.8240 |
| | 24 | **0.846 (0.004)** | **0.803 (0.005)** | **0.8245** |
| | 32 | 0.844 (0.005) | 0.802 (0.006) | 0.8230 |

In fact, $\widehat{l}$ and $\breve{l}$ can be set in a wide range, as lengths of core fragments vary over a wide range. Specifically, if the core fragment is spread across multiple fragments, features of this core fragment can be further extracted by the 2D CNN. If the core fragment is too short, features of this core fragment can be further extracted by weighting features of the corresponding fragment, which also can be extracted by 2D CNN. Furthermore, the RoBERTa needs to be trained on large-scale data, so $\widehat{l}$ and $\breve{l}$ are hardly set by the experiment. As a result, $\widehat{l}$ and $\breve{l}$ are simply set to 16 and 128 according to above analyses but not according to the experiment.

Next, the principle of setting $K$ is introduced. $K$ should be set to the value that most words are distributed in one or more fragments. Therefore, $K * \widehat{l}$ and $K * \breve{l}$ should be respectively larger than 81 and 1,732, which are the upper limit of most drugs and targets presented in Figs. 1 and 2. Furthermore, some overlap should exist among the neighbor fragments. As a result, $K$ should be larger than 13.5 which is 1,732/128, and results of ASSLDTA whose $K$ is set to 16, 24, 32 are presented in Table 5, where the best of each group on each dataset is shown in bold.

It can be seen from Table 5 that CIs, MSE and AUPR of ASSLDTA are the best when $K$ is set to 24, and $r_m^2$ of ASSLDTA are the best on most datasets when $K$ is set to 16. Furthermore, results of ASSLDTA when $K$ is set to 32 are much worse than results of ASSLDTA when $K$ is set to 16 and 24, the reason may be that low-level features extracted by ASSL contain too much positional information when $K$ is too large. As a result, $K$ is set to 24 in the next experiments, as $K$ set to 24 is the balance for the number of the fragments and the positional information. Additionally, Table 5 also indicates that adjusting $K$ within a wide range does not significantly impact the results. The primary reason is that $K$ only affects the degree of window overlap, and the subsequent 2D CNN can adapt to different degrees of window overlap to a certain extent. We set the same $K$ value for all databases, which to some degree demonstrates the good generalization ability of ASSLDTA.

## Results of DTA on Kiba

In this subsection, ASSLDTA is compared with many recent works on Kiba, such as DeepDTA (*Öztürk, Özgür & Ozkirimli, 2018*), MATT (*Zeng et al., 2021*), WideDTA (*Öztürk, Ozkirimli & Özgür, 2019*), MAM (*Aleb, 2021*), AttentionDTA (*Zhao et al., 2019*), SAG-DTA (*Zhang et al., 2021*), GraphDTA (*Nguyen, Le & Venkatesh, 2021*), MGraphDTA (*Pan et al., 2023*), DeepGS (*Lin, 2020*), DGraphDTA (*Chen et al., 2020*), GLCNDTA (*Gao et al., 2018*), TDGraphDTA (*Zhu et al., 2023a*), DeepGLSTM (*Mukherjee, Ghosh & Basuchowdhuri, 2022*), GCN-BERT (*Lennox, Robertson & Devereux, 2021*), and DGDTA (*Yang et al., 2022*). To comprehensively consider the role of each module in ASSLDTA, ASSLDTA is also compared with MAM, 2DCNN-GCN, 2DCNN-BERT, MPBERT and CPBERT, where the backbone network of MAM is 2DCNN, the network structure of 2DCNN-GCN is similar with ASSLDTA but its input is one-hot vector, the input of 2DCNN-BERT is consistent with ASSLDTA but it does not have GCN. MPBERT employees Mol-BERT (*Xia et al., 2023b*) to extract features for drugs, while ProteinBERT (*Brandes et al., 2022*) is utilized to capture features for targets. CPBERT employees ChemBERTa-2 (*Ahmad et al., 2022*) to extract features for drugs, while ProteinBERT (*Brandes et al., 2022*) is utilized to capture features for targets. The model evaluations of the DTA on Kiba are listed in Table 6, where the best of each group on each dataset is shown in the bold font. It can be seen from Table 6 that ASSLDTA is the best among all compared methods on Kiba.

Firstly, ASSLDTA is much better than all CNN-based methods. The CI, MSE, $r_m^2$ and AUPR of ASSLDTA are much higher than that of the compared CNN-based methods. Specifically, the CI of ASSLDTA is 0.01 higher than the CI of MAM, and the MSE of ASSLDTA is 0.011 less than the MSE of MAM, where MAM is the best method among CNN-based methods. They prove that using ASSL to learn low-level features for drugs and targets is better for DTAP.

Secondly, ASSLDTA is much better than all GCN- and CNN-based methods. The CI, MSE, $r_m^2$ and AUPR of ASSLDTA are much higher than that of the compared GCN- and CNN-based methods. Specifically, the CI of ASSLDTA is 0.004 higher than the CI of DGraphDTA, and the MSE of ASSLDTA is 0.002 less than the MSE of DGraphDTA, where DGraphDTA is the best method among GCN- and CNN-based methods. They prove that using ASSL together with 2D CNN to learn high-level features is better.

Thirdly, ASSLDTA is much better than other pre-trained model-based methods. There are seven pre-trained models for comparison, including DeepGLSTM, GCN-BERT, SubMDTA, TransVAEDTA, 2DCNN-BERT, MPBERT, and CPBERT. The CI, MSE, $r_m^2$ and AUPR of ASSLDTA are much higher than that of other pre-training model-based methods. Specifically, the CI, MSE, $r_m^2$ and AUPR of ASSLDTA are 0.004, −0.004, 0.019 and 0.016 higher than the CI of CPBERT, where CPBERT is the best method among the pre-trained model-based methods by using different pre-training methods. They prove that the effectiveness of our improved RoBERTa for SMILES data and protein sequences compared to other architectures. The possible reason is as follows: other methods use the entire drug and target as training data, which may obscure the fragment features that play a

**Table 6 Results of DTA on Kiba.**

| Method | Drug | Target | CI↑ | MSE↓ | $r_m^2$↑ | AUPR↑ |
|---|---|---|---|---|---|---|
| **CNN-based method** | | | | | | |
| DeepDTA (*Öztürk, Özgür & Ozkirimli, 2018*) | 1D-CNN | 1D-CNN | 0.863 (0.002) | 0.194 | 0.673 (0.009) | 0.788 (0.004) |
| MATT (*Zeng et al., 2021*) | 1D-CNN | 1D-CNN | 0.889 (0.004) | 0.150 | 0.756 (0.011) | – |
| WideDTA (*Öztürk, Ozkirimli & Özgür, 2019*) | 1D-CNN | 1D-CNN | 0.875 (0.001) | 0.179 (0.008) | 0.692 (0.009) | – |
| AttentionDTA (*Zhao et al., 2019*) | 1D-CNN | 1D-CNN | 0.882 (0.004) | 0.162 (0.003) | 0.755 (0.017) | 0.829 (0.005) |
| **GCN- and CNN-based method** | | | | | | |
| SAG-DTA (*Zhang et al., 2021*) | GCN | 1D-CNN | 0.892 | 0.130 | | |
| GraphDTA (*Nguyen, Le & Venkatesh, 2021*) | GCN | 1D-CNN | 0.891 | 0.139 | 0.736 (0.028) | 0.823 (0.009) |
| MGraphDTA (*Yang et al., 2022*) | GCN | 2D-CNN | 0.902 (0.001) | 0.128 (0.001) | – | – |
| DeepGS (*Lin, 2020*) | GCN, BiGRU | 2D-CNN | 0.860 | 0.193 | – | – |
| DGraphDTA (*Chen et al., 2020*) | GCN | GCN | 0.904 | 0.126 | - | - |
| GLCNDTA (*Gao et al., 2018*) | GCN | GCN | 0.899 | 0.127 | – | – |
| TDGraphDTA (*Zhu et al., 2023a*) | GCN | CNN | 0.899 | 0.121 | – | – |
| **Pre-training model based method** | | | | | | |
| DeepGLSTM (*Mukherjee, Ghosh & Basuchowdhuri, 2022*) | GCN | LSTM | 0.897 | 0.133 | – | – |
| GCN-BERT (*Lennox, Robertson & Devereux, 2021*) | GCN | BERT | 0.888 (0.001) | 0.149 (0.001) | 0.761 (0.009) | 0.838 (0.003) |
| SubMDTA (*Pan et al., 2023*) | Pre-trained GIN | BiLSTM | 0.898 | 0.129 | 0.793 | |
| TransVAEDTA (*Zhou et al., 2024*) | VAE | Transformer | 0.822 (0.002) | 0.253 | 0.632 (0.001) | 0.701 (0.004) |
| **Ablation method** | | | | | | |
| MAM (*Aleb, 2021*) | 2D-CNN | 2D-CNN | 0.898 | 0.135 | – | – |
| 2D-CNN-GCN | GCN+2D-CNN | 2D-CNN | 0.902 (0.003) | 0.131 (0.003) | 0.752 (0.007) | 0.826 (0.005) |
| 2DCNN-BERT | BERT+2D-CNN | BERT+2D-CNN | 0.907 (0.002) | 0.125 (0.002) | 0.780 (0.005) | 0.841 (0.005) |
| MPBERT | Mol-BERT | ProteinBERT | 0.903 (0.003) | 0.130 (0.003) | 0.769 (0.006) | 0.825 (0.005) |
| CPBERT | ChemBERTa-2 | ProteinBERT | 0.904 (0.002) | 0.128 (0.003) | 0.766 (0.005) | 0.830 (0.004) |
| **ASSLDTA** | **GCN, BERT+2D-CNN** | **BERT+2D-CNN** | **0.908 (0.002)** | **0.124 (0.002)** | **0.785 (0.005)** | **0.846 (0.004)** |

**Note:**
Values in bold represent the results of the proposed method.

crucial role in drug-target affinity. In contrast, our pre-training approach focuses on feature extraction from fragments, enabling better preservation of local fragment features favorable to drug-target affinity, thus facilitating the subsequent application of supervised methods for highlighting these features.

Fourthly, the module of BERT is the most important in ASSLDTA. It can be seen from Table 6 that the CI of ASSLDTA is respectively 0.010, 0.006, 0.01 0.005 and higher than that of 2DCNN-BERT, 2D-CNN-GCN, MAM and GBERT. The MSE of ASSLDTA is 0.002, 0.008, 0.012 and 0.006 less than that of 2DCNN-BERT, 2D-CNN-GCN and MAM. Furthermore, $r_m^2$ and AUPR of ASSLDTA are also much higher than these ablation methods. They demonstrate that ASSL can significantly enhance the performance of DTAP, while GCN has limited impact on enhancing DTAP.

## Results of DTA on Davis

In this subsection, ASSLDTA is compared with many recent works on Davis, such as DeepDTA (*Öztürk, Özgür & Ozkirimli, 2018*), MATT (*Zeng et al., 2021*), WideDTA (*Öztürk, Ozkirimli & Özgür, 2019*), MAM (*Aleb, 2021*), AttentionDTA (*Zhao et al., 2019*), SAG-DTA (*Zhang et al., 2021*), GraphDTA (*Nguyen, Le & Venkatesh, 2021*), MGraphDTA (*Pan et al., 2023*), DeepGS (*Lin, 2020*), DGraphDTA (*Chen et al., 2020*), GLCNDTA (*Gao et al., 2018*), TDGraphDTA (*Zhu et al., 2023a*), DeepGLSTM (*Mukherjee, Ghosh & Basuchowdhuri, 2022*), GCN-BERT (*Lennox, Robertson & Devereux, 2021*), and DGDTA (*Yang et al., 2022*). To comprehensively consider the role of each module in ASSLDTA, ASSLDTA is also compared with MAM, 2DCNN-GCN, 2DCNN-BERT, MPBERT and CPBERT, where the backbone network of MAM is 2DCNN, the network structure of 2DCNN-GCN is similar with ASSLDTA but its input is one-hot vector, the input of 2DCNN-BERT is consistent with ASSLDTA but it does not have the GCN. GBERT employees Mol-BERT (*Xia et al., 2023b*) to extract features for drugs, while ProteinBERT (*Brandes et al., 2022*) is utilized to capture features for targets. CPBERT employees ChemBERTa-2 (*Ahmad et al., 2022*) to extract features for drugs, while ProteinBERT (*Brandes et al., 2022*) is utilized to capture features for targets. The model evaluations of the DTA on Davis are listed in Table 7, where Davis was often used to evaluate DTAP. It can be seen from Table 6 that ASSLDTA is the best among all compared methods on Davis.

Firstly, ASSLDTA is much better than all CNN-based methods. The CI, MSE, $r_m^2$ and AUPR of ASSLDTA are much higher than that of the compared CNN-based methods. Specifically, the CI of ASSLDTA is 0.017 higher than the CI of AttentionDTA, and the MSE of ASSLDTA is 0.017 less than the MSE of AttentionDTA, where AttentionDTA is the best method among CNN-based methods. They prove that using ASSL to learn low-level features for drugs and targets is better for DTAP.

Secondly, ASSLDTA is much better than all GCN- and CNN-based methods. The CI, MSE, $r_m^2$ and AUPR of ASSLDTA are much higher than that of the compared GCN- and CNN-based methods. Specifically, the CI of ASSLDTA is 0.004 higher than the CI of TDGraphDTA, where TDGraphDTA is the best method among GCN- and CNN-based methods. They prove that using ASSL together with 2D CNN to learn high-level features is better.

Thirdly, ASSLDTA is much better than other pre-trained model-based methods. There are seven pre-trained models for comparison, including DeepGLSTM, GCN-BERT, SubMDTA, TransVAEDTA, 2DCNN-BERT, MPBERT, and CPBERT. The CI, MSE, $r_m^2$ and AUPR of ASSLDTA are much higher than that of other pre-training model-based methods. Specifically, the CI, MSE, $r_m^2$ and AUPR of ASSLDTA are 0.007, −0.002, 0.009 and 0.008 higher than the CI of CPBERT, where CPBERT is the best method among the pre-trained model-based methods by using different pre-training methods. They prove that the effectiveness of our improved RoBERTa for SMILES data and protein sequences compared to other architectures. The possible reason is that other methods use the entire drug and target as training data, which may obscure the fragment features that play a

**Table 7  Results of DTA on Davis.**

| Method | Drug | Target | CI↑ | MSE↓ | $r_m^2$↑ | AUPR↑ |
|---|---|---|---|---|---|---|
| **CNN-based method** | | | | | | |
| DeepDTA (*Öztürk, Özgür & Ozkirimli, 2018*) | 1D-CNN | 1D-CNN | 0.878 (0.004) | 0.261 | 0.630 (0.017) | 0.714 (0.010) |
| MATT (*Zeng et al., 2021*) | 1D-CNN | 1D-CNN | 0.891 (0.003) | 0.227 | 0.683 (0.009) | – |
| WideDTA (*Öztürk, Ozkirimli & Özgür, 2019*) | 1D-CNN | 1D-CNN | 0.886 (0.003) | 0.262 (0.009) | 0.633 (0.007) | – |
| AttentionDTA (*Zhao et al., 2019*) | 1D-CNN | 1D-CNN | 0.893 (0.005) | 0.216 | 0.677 (0.024) | 0.776 (0.024) |
| **GCN- and CNN-based method** | | | | | | |
| SAG-DTA (*Zhang et al., 2021*) | GCN | 1D-CNN | 0.903 | 0.209 | | |
| GraphDTA (*Nguyen, Le & Venkatesh, 2021*) | GCN | 1D-CNN | 0.893 | 0.229 | 0.656 (0.014) | 0.710 (0.006) |
| MGraphDTA (*Yang et al., 2022*) | GCN | 2D-CNN | 0.900 (0.004) | 0.207 (0.001) | – | – |
| DeepGS (*Lin, 2020*) | GCN, BiGRU | 2D-CNN | 0.882 | 0.252 | – | – |
| DGraphDTA (*Zhang et al., 2019*) | GCN | GCN | 0.904 | 0.202 | – | - |
| GLCNDTA (*Gao et al., 2018*) | GCN | GCN | 0.903 | 0.215 | – | – |
| TDGraphDTA (*Zhu et al., 2023a*) | GCN | CNN | 0.906 | 0.199 | – | – |
| **Pre-training model based method** | | | | | | |
| DeepGLSTM (*Mukherjee, Ghosh & Basuchowdhuri, 2022*) | GCN | LSTM | 0.895 | 0.232 | – | – |
| GCN-BERT (*Lennox, Robertson & Devereux, 2021*) | GCN | BERT | 0.896 (0.002) | 0.199 (0.003) | 0.741 (0.002) | 0.806 (0.007) |
| SubMDTA (*Pan et al., 2023*) | Pre-trained GIN | BiLSTM | 0.894 | 0.218 | 0.719 | - |
| TransVAEDTA (*Zhou et al., 2024*) | VAE | Transformer | 0.869 (0.008) | 0.332 | 0.571 (0.001) | 0.662 (0.003) |
| **Ablation method** | | | | | | |
| MAM (*Aleb, 2021*) | 2D-CNN | 2D-CNN | 0.891 | 0.183 | - | - |
| 2D-CNN-GCN | GCN+2D-CNN | 2D-CNN | 0.904 (0.003) | 0.206 (0.004) | 0.735 (0.006) | 0.788 (0.005) |
| 2DCNN-BERT | BERT+2D-CNN | BERT+2D-CNN | 0.908 (0.002) | 0.201 (0.003) | 0.752 (0.003) | 0.801 (0.005) |
| MPBERT | Mol-BERT | ProteinBERT | 0.902 (0.004) | 0.202 (0.003) | 0.746 (0.003) | 0.793 (0.004) |
| CPBERT | ChemBERTa-2 | ProteinBERT | 0.903 (0.003) | 0.201 (0.003) | 0.749 (0.003) | 0.795 (0.004) |
| **ASSLDTA** | **GCN, BERT+2D-CNN** | **BERT+2D-CNN** | **0.910 (0.002)** | **0.199 (0.002)** | **0.758 (0.004)** | **0.803 (0.005)** |

**Note:**
Values in bold represent the results of the proposed method.

crucial role in drug-target affinity. In contrast, our pre-training approach focuses on feature extraction from fragments, enabling better preservation of local fragment features favorable to drug-target affinity, thus facilitating the subsequent application of supervised methods for highlighting these features.

Fourthly, the module of BERT is the most important in ASSLDTA. It can be seen from Table 7 that the CI of ASSLDTA is respectively 0.002, 0.006, 0.019 and 0.008 higher than that of 2DCNN-BERT, 2DCNN-GCN, MAM and GBERT. The MSE of ASSLDTA is 0.002, 0.007 and 0.003 less than that of 2DCNN-BERT, 2DCNN-GCN and GBERT. They demonstrate that ASSL can significantly enhance the performance of DTAP, while GCN has limited impact on enhancing DTAP.

Furthermore, although the MSEs of GCN-BERT and MAM are better than that of ASSLDTA on Davis, MSEs, the CIs of GCN-BERT and MAM are much worse than those

of ASSLDTA on Kiba, and the CIs of GCN-BERT and MAM are much worse than those of ASSLDTA on Davis. They prove that GCN-BERT and MAM are not stable and ASSLDTA are also much better than GCN-BERT and MAM.

## Results of DTA on other three datasets

In this subsection, ASSLDTA is compared with GraphDTA and DeepGLSTM on DTC, Metz, and Tox-cast. Only GraphDTA and DeepGLSTM are compared, as few deep methods are currently validated on these three databases. Results are presented in Figs. 5 and 6.

It can be seen from Fig. 5 and 6 that our method is also the best. Firstly, the CI of ASSLDTA is respectively 0.025, 0.015, and 0.008 higher than that of GraphDTA, and the MSE of ASSLDTA is respectively 0.034, 0.038, and 0.027 less than that of GraphDTA on DTC, Metz, and Tox-Cast. Secondly, the CI of ASSLDTA is respectively 0.006, 0.005, and 0.004 higher than that of GraphDTA and the MSE of ASSLDTA is respectively 0.007, 0.015, and 0.016 less than that of GraphDTA on DTC, Metz, and Tox-Cast. They prove that using ASSL together with 2D CNN to learn high-level features for the drug and target is better for DTAP and demonstrates the good generalization ability of ASSLDTA.

## Results of CPI

In this subsection, the binary classification task of CPI is also used to evaluate the ASSLDTA. ASSLDTA is compared with MGraphDTA (*Wang et al., 2024*), GraphDTA (*Nguyen, Le & Venkatesh, 2021*), SAG-DTA (*Zhang et al., 2021*), SVM (*Chen et al., 2020*), TransformerCPI (*Chen et al., 2020*), GNN-CNN (*Zhao et al., 2024*), and TrimNet (*Li et al., 2021*). The results are presented in Table 8, where our method is shown in bold.

It can be seen from Table 8 that AUROC of ASSLDTA is the best on all datasets. AUROCs of ASSLDTA are respectively 0.005, 0.003 and 0.006 higher than that of SAG-DTA, MGNN-MCNN and SAG-DTA on the Human, C.elegans and BindingDB datasets, where SAG-DTA, MGraphDTA and SAG-DTA are respectively the second best methods on these three datasets. They prove that ASSLDTA is also benefit for CPI, which also prove the good generalization ability of ASSLDTA.

## Fill mask evaluations of the ASSL

In this subsection, the filling mask task on five DTA datasets is used to evaluate the effectiveness of ASSL. In the padding mask task, the second to last character of a SMILES or a sequence is masked sequentially. The RoBERTa model is then used to predict the masked characters. The more accurate the prediction of the character, the better the model can extract the features of the drug or target. The predicted precisions of the characters are presented in Figs. 7 and 8, where the characters around the perimeter have been sorted according to the top-1 precisions. The overall prediction precisions for the masked characters are presented in Figs. 9 and 10. Top-p indicates that the masked character is in the set of characters with top-p prediction scores.

ASSL trained on drug fragments has a strong ability to reconstruct drug fragments. It can be seen from Fig. 7 that the Top-3 precisions of most characters are greater than 0.6.

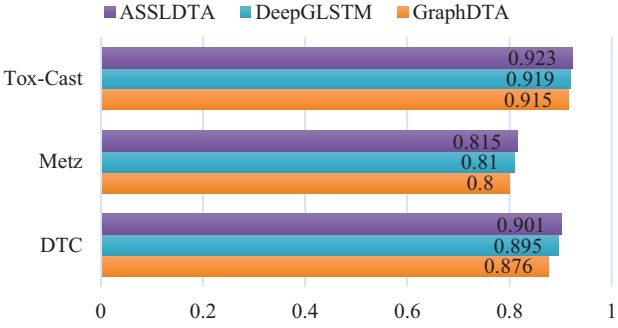

**Figure 5 CIs of DTA on DTC, Metz and Tox-Cast.**

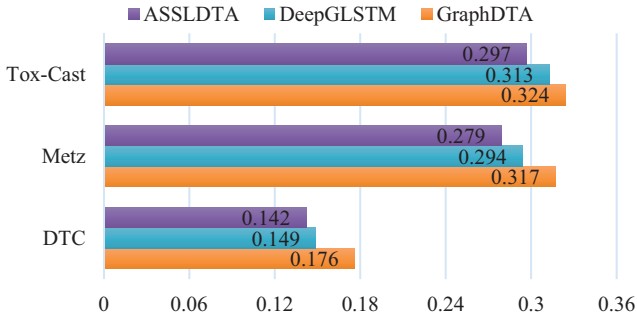

**Figure 6 MSEs of DTA on DTC, Metz and Tox-Cast.**

**Table 8 Results of CPI.**

| Method | Human | C.elegans | BindingDB |
|---|---|---|---|
| MGraphDTA (*Wang et al., 2024*) | 0.983 | 0.991 | – |
| GraphDTA (*Nguyen, Le & Venkatesh, 2021*) | 0.960 | 0.974 | 0.929 |
| SAG-DTA (*Zhang et al., 2021*) | 0.984 | – | 0.963 |
| SVM (*Chen et al., 2020*) | 0.910 | 0.894 | – |
| TransformerCPI (*Chen et al., 2020*) | 0.973 | 0.988 | 0.951 |
| GNN-CNN (*Zhao et al., 2024*) | 0.970 | 0.978 | – |
| TrimNet (*Li et al., 2021*) | 0.974 | 0.987 | – |
| **ASSLDTA** | **0.989** | **0.994** | **0.969** |

Note:
The bold represents the results of our method.

Specifically, the Top-10 precisions of most characters are greater than 0.95. It can be seen from Fig. 9 that the overall Top-1, Top-3, Top-5 and Top-10 precisions of the SMILES are respectively larger than 0.624, 0.855, 0.917, and 0.967, where the number of different characters is greater than 28. They support the validity of the features extracted by ASSL for drugs.

The ASSL trained on fragments of the target also has a strong ability to reconstruct the fragment of the target. It can be seen from Fig. 8 that the Top-1 precisions of most characters are greater than 0.4. Specifically, the Top-10 precisions of most characters are greater than 0.80. It can be seen from Fig. 10 that the overall Top-1, Top-3, Top-5, and

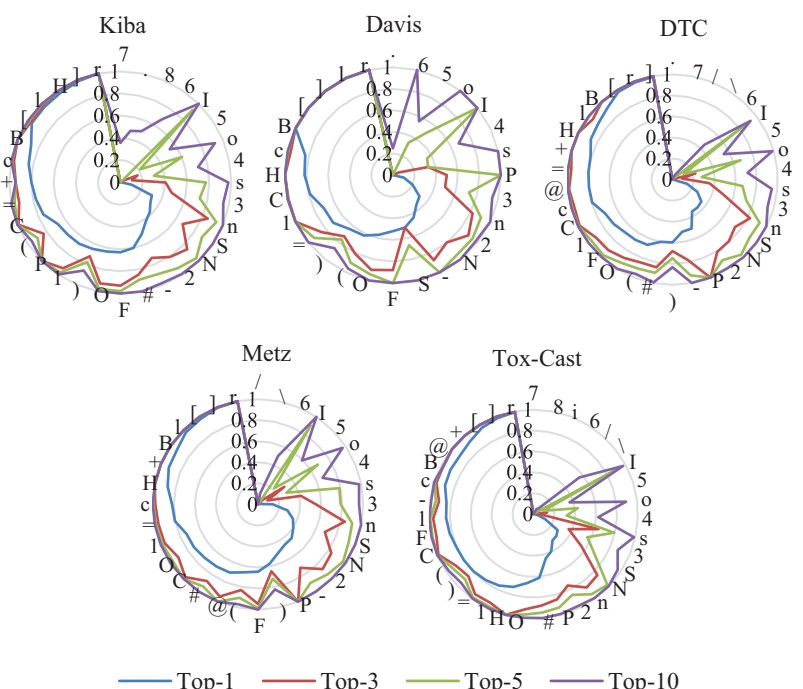

**Figure 7  Prediction precisions of each character of SMILES on five DTA datasets.**

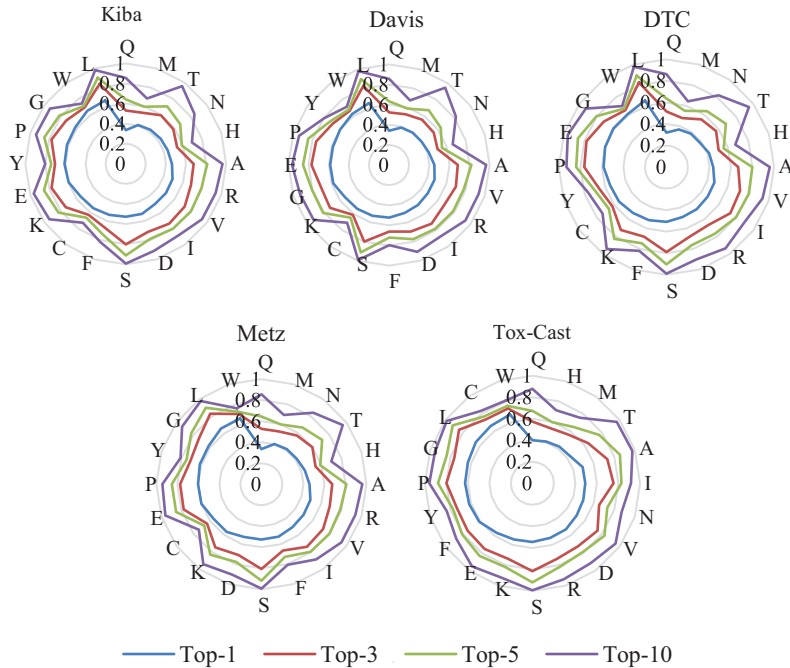

**Figure 8  Prediction precisions of each character of sequences on five DTA datasets.**

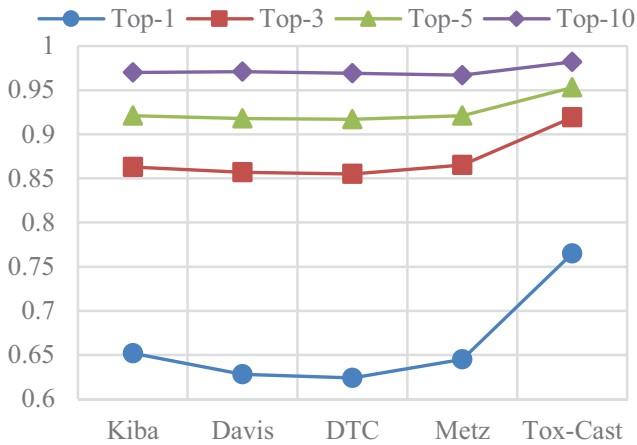

**Figure 9 Mean prediction precisions of characters of SMILES on five DTA datasets.**

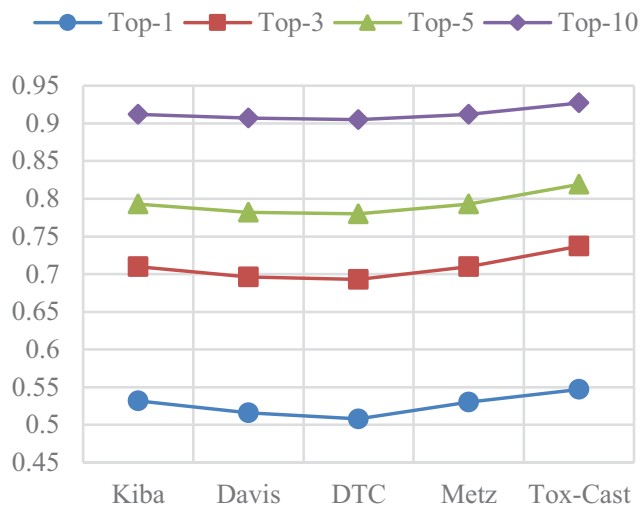

**Figure 10 Mean prediction precisions of characters of sequences on five DTA datasets.**

Top-10 precisions of the sequence are respectively larger than 0.508, 0.696, 0.780, and 0.905, where the number of different characters is greater than 20. They support the effectiveness of the features of the target extracted by the ASSL.

Furthermore, it can be seen from Figs. 7–10 that the prediction precisions of SMILESs are better than that of the sequence. However, in ASSLDTA, GCN is also used to extract features for SMILES. The reasons are as follows: Firstly, the Top-1 precisions of many characters are very low, which may have a negative impact on the quality of drug feature extraction. It can be seen from Fig. 7 that the Top-1 precisions of about 1/3 characters are smaller than 0.4, such as '.', '\', '/', '7', '8', '6', ')', '5' and '4'. Secondly, the molecular structure of a drug is a typical graph structure, and then effective drug features can be extracted by using a GCN.

## DISCUSSION

Pre-training typically involves using unsupervised learning techniques to train the model on a large amount of unlabeled data, which has been successfully applied in many fields. However, there are still some problems with directly applying pre-training to DTAP. A drug and a target should be provided for DTAP, but there is no way to provide sufficient drug target pairs for pre-training. One optional approach is to pre-train separately for drugs and targets. However, from the principle of drug design, the DTAP is mainly determined by core fragments of the SMILES and sequence (*Lin, Zhang & Xu, 2020*; *Zhang et al., 2019*). As a result, it is difficult to improve the DTAP effect by directly utilizing the high-level features learned through separate pre-training strategies (*Liu et al., 2019*; *Lennox, Robertson & Devereux, 2021*; *Elnaggar et al., 2020*; *Liu et al., 2021*).

In order to better apply pre-trained models to DTAP, we consider first using pre-trained models to learn the features of drug and target fragments, and then using 2D CNN to further learn high-level features in a supervised manner. Thus, guided by the label, 2DCNN does not need to learn the features of the fragments from scratch, but only needs to highlight the features of the core fragments, which is necessary in DTAP where the number of training samples of DTAP is still very small.

Our experiments and results are validated and reflected the effectiveness of ASSLDTA. ASSLDTA performs much better than CNN-based methods, suggesting that using ASSL to prepare better inputs for CNN can improve the performance of DTA. ASSLDTA performs much better than GCN and CNN-based methods, showing that ASSL together with 2D CNN can learn effective drug features and target features. ASSLDTA performs much better than GCN and pre-trained model-based methods, indicating that it is important to improve SSL based on the characteristics of DTA. ASSLDTA is not parameter sensitive and ASSLDTA can improve the effectiveness of CPI, which indicates that ASSLDTA has good generalizability of our method and can be easily used in drug screening.

Although our method has a better overall performance than the compared method, its improvement is limited. This is because there are still some limitations in our research work. For example, SSL is utilized to extract low-level features but not directly improved to overcome the problems of existing SSL methods. DTA is determined by core fragments of drugs and targets, but we did not directly design a method that can precisely divide the core fragments. These works open room for future investigations. One of the further works will be to design a new training goal for pre-training to directly overcome the problems of existing pre-training methods. As another further work, a better segmentation method will be designed to allow each fragment to have a specific function, and then the features extracted from each fragment by the pre-training would be more meaningful.

## CONCLUSION

In this article, we propose an ASSLDTA for DTA prediction that can leverage a large number of unlabeled segments to learn low-level features by ASSL and further leverage a small number of labeled samples to learn high-level features by 2DCNN. In particular, we have changed the way of using pre-trained models in DTAP, which can overcome three

problems of commonly usage strategy, such as the gap between the input of pre-training and DTAP, the gap between the output of pre-training and the need of DTAP, and the gap between training goals of pre-training and DTAP. In addition, GCN is also used to extract graph features, where drug molecules have typical graph structures. Experiments have been conducted to verify that ASSLDTA is a competitive method compared to previous methods.

### Funding
This work was supported by the, National Natural Science Foundation of China (No. 61972299), the Zhejiang "Lingyan" Research and Development Program (No. 2022C03121), and the Wenzhou Natural Science Foundation (No. 2021G0170). The funders had no role in study design, data collection and analysis, decision to publish, or preparation of the manuscript.

### Grant Disclosures
The following grant information was disclosed by the authors:
National Natural Science Foundation of China: 61972299.
Zhejiang "Lingyan" Research and Development Program: 2022C03121.
Wenzhou Natural Science Foundation: 2021G0170.

### Competing Interests
Yaxin Sun is employed by Zhejiang Aerospace Hengjia Data Technology Co. Ltd.

### Author Contributions
- Qing Ye conceived and designed the experiments, performed the experiments, analyzed the data, performed the computation work, prepared figures and/or tables, authored or reviewed drafts of the article, and approved the final draft.
- Yaxin Sun conceived and designed the experiments, authored or reviewed drafts of the article, and approved the final draft.

### Data Availability
The Kiba, Davis, DTC, Metz and Tox-Cast datasets are available at GitHub: https://github.com/MLlab4CS/DeepGLSTM.
The Human, C.elegans, and BindingDB datasets are available at GitHub: https://github.com/lifanchen-simm/transformerCPI.
The ASSLDTA code is available in the Supplemental File and at GitHub: https://github.com/yeqing0713/ASSLDTA.

### Supplemental Information
Supplemental information for this article can be found online at http://dx.doi.org/10.7717/peerj-cs.2622#supplemental-information.

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
