# Peer review of "Improving drug–target affinity prediction by adaptive self-supervised learning"

_PeerJ Computer Science, doi:10.7717/peerj-cs.2622_

## Round 0.1 · original submission · Major Revisions

Reviewers found that your manuscript is currently not ready to be accepted for publication at PeerJ Computer Science. However, we would like you to revise your work. Please read the reviewer's comments and revise your manuscript following these points.

·

Basic reporting

The manuscript is well-written and provides a good overview of the DTA problem, the limitations of existing methods, and the proposed ASSL approach. The methodology section describes ASSLDTA components, including ASSL training, feature extraction, and DTA score learning. However, some areas could benefit from clarification:

1. Elaborate on the novelty of ASSL: While the authors mention that ASSL is based on ROBERTa, a more detailed explanation of the modifications and their impact on addressing the specific challenges of DTA would strengthen the manuscript.
2. Provide more insights into the architecture choices: The rationale behind using 2D CNN and GCN could be further elaborated.
3. Enhance the visual presentation: Figures 1 and 2 could benefit from improved clarity and labeling for better readability.

Experimental design

The experimental design is robust. The authors assess ASSLDTA using multiple datasets and compare its performance with various state-of-the-art methods. The ablation study offers insights into the contributions of different modules within ASSLDTA. However, there are certain aspects of the experimental design that could be improved:

1. Justify the choice of hyperparameters: The selection of hyperparameters, especially for ASSL, lacks sufficient justification.
2. Consider additional evaluation metrics: Including metrics beyond CI and MSE, such as those focusing on early recognition, would provide a more comprehensive performance assessment.
3. Compare with other pre-training strategies: While the authors compare ASSLDTA with DeepGLSTM and GCN-BERT, exploring additional recent pre-training techniques would further validate the effectiveness of ASSL.

Validity of the findings

The results support the authors' claims. ASSLDTA outperforms other methods, indicating the effectiveness of combining ASSL and 2D CNN for DTA prediction. However, concerns remain:

1. Address the limitations of ASSL: it relies on fixed-length fragments and does not explicitly address core fragment identification.
2. Discuss the generalizability of ASSLDTA: further investigations are needed to confirm its robustness and generalizability to other DTA prediction tasks and datasets.

Reviewer 2 ·

Basic reporting

The authors have proposed a method for predicting the affinity between drugs and targets based on Adaptive Self-Supervised Learning for Drug-Target Affinity (ASSLDTA). In ASSLDTA, features are extracted through a newly designed adaptive self-supervised learning method and a 2D convolutional neural network. Adaptive self-supervised learning can learn the features of drug and target segments using a large amount of unlabeled training samples. The 2D convolutional neural network can learn the features among segments using a small number of labeled training samples. The authors also tested their method on 5 DTA datasets and 3 CPI datasets. The manuscript is well-written with professional English. The manuscript content is indicative with clear information. However, I have some concerns about this work, as follows.

Experimental design

Section 1. Introduction
- The authors mentioned that “In CNN, the convolution kernel can extract features for a local window, which can be viewed as a fragment of the input. However, the size of the convolution kernel cannot be too big, which makes that the fragment divided by CNN is insufficient to describe the core fragment.” On which basis is this sentense based? Citations must be provided to clarify this point. In my opinion, changing the window size can play roles as hyperparameters in modeling and these hyperparameters are tunable and customizable.

- Another problem is that the high-level feature is directly extracted from raw input, which could be hard in DTAP, as the labeled training samples are too limited”. I think that the small size of the training data set is a common issue for supervised models, not just for GCN
SubSection 2.1 Data
- The authors need to add information about ratio of posotive and negative of 5 DTA datasets
SubSection 2.2 Problems of SSL usd in DTAP and our Solution strategy
- When describing the SSL method, the authors only mentioned RoBERT (A model used for text). In this section, I recommend that authors should compare theirs to other SSL models designed for molecules (e.g., MolRoPE-BERT or ProteinBERT). These models has been trained on a vast number of drug molecules and protein, enhacing feature extraction of significant information. Authors can re-implement these models to compare and evaluate the performnace on their datasets.

SubSection 2.3 ASSLDTA structure
- The proposed architecture by the author is not very remarkable; the features taken from ASSL are essentially just features of RoBERTa for sequence data.
- The branch that uses GCN in the author's proposed architecture needs to describe how the data encoding process is carried out.
- Apart from concatenating high-level features, the authors have tried other methods such as attention or matrix multiplication.

SubSection 3.2 Parametric analysis of the ASSLDTA
- The experiment tuning K and the results of group tuning should be run on the validation set of the datasets.

Validity of the findings

This work lacks statistical evidence to support the reproducibility and robustness of proposed method. Please provide it.

Cite this review as

---

## Round 0.2 · Minor Revisions

Based on the reviewers' comments, we have found that your work needs to be further improved. We are happy to offer you an opportunity to revise your work. Please check and address all points raised by reviewers.

·

Basic reporting

- The manuscript is well-organized, with improved explanations of ASSL and its role in DTA prediction.
- Figures 1 and 2 have been revised for better clarity.
- The manuscript addressed all the previous concerns and meets the journal's scope.

Experimental design

- Hyperparameter selection is clearly explained, fitting the dataset and model complexity.
- The experiments evaluated with state-of-the-art methods and multiple datasets confirm the robustness of the proposed model.
- The experimental design is enough to meet the journal's standard.

Validity of the findings

- The model has proven that it can work well across different datasets, showing that it is efficient and reliable.
- The authors discussed the issue of fixed-length fragments and explained how their approach solves this problem.
- The findings are well-supported and contribute to the field, meeting all the requirements for publication.

Reviewer 2 ·

Basic reporting

- The authors designed a two-stage feature extraction method, which includes an ASSL and a 2D CNN. The ASSL can learn enough information to reconstruct fragments and describe the relationships among them and their neighbors using unlabeled samples. The 2D CNN can further highlight useful fragments and their neighbor relationships with a small number of labeled training samples.
- The authors evaluated the proposed model on five benchmark datasets for drug-target affinity prediction (DTAP), namely Kiba, Davis, DTC, Metz, and Tox-Cast. The proposed model was also evaluated on three benchmark binary classification datasets for compound–protein interaction (CPI) prediction, specifically BindingDB, C. elegans, and Human.
- Figure 4 shows the ratio and position of non-uniform blocks.
- In the proposed architecture, the authors use a GCN architecture. They need to describe the process of encoding the dataset, specifically how they encoded SMILES into molecular graphs and what features were used.
- The authors used RoBERTa, an architecture designed for text processing. However, in this study, we are dealing with data for molecules and proteins. The authors need to demonstrate the effectiveness of RoBERTa for SMILES data and protein sequences compared to other architectures. Currently, there are several pre-trained BERT models customized specifically for SMILES, such as ChemBERT and SMILES-BERT. Similarly, for proteins, there are pre-trained BERT models like ProteinBERT. The authors should conduct further experiments and discussions

Experimental design

- In the proposed architecture, the authors use a GCN architecture. They need to describe the process of encoding the dataset, specifically how they encoded SMILES into molecular graphs and what features were used.
- The authors used RoBERTa, an architecture designed for text processing. However, in this study, we are dealing with data for molecules and proteins. The authors need to demonstrate the effectiveness of RoBERTa for SMILES data and protein sequences compared to other architectures. Currently, there are several pre-trained BERT models customized specifically for SMILES, such as ChemBERT and SMILES-BERT. Similarly, for proteins, there are pre-trained BERT models like ProteinBERT. The authors should conduct further experiments and discussions.

Validity of the findings

- How did the authors handle the data splitting for each dataset? Basic statistics are needed on the number of samples in the training, validation, and test sets for each dataset.

Cite this review as

---

## Round 0.3 · accepted · Accept

Based on the reviewer's comments, we are pleased to inform you that your work has been accepted for publication at PeerJ Computer Science.

Reviewer 2 ·

Basic reporting

The authors have considered all the comments.

Experimental design

This version is ready for publication.

Validity of the findings

The findings presented in the manuscript are valid, well-supported by the data, and align with the study's objectives

Cite this review as